# PSMA PET Imaging and Therapy in Adenoid Cystic Carcinoma and Other Salivary Gland Cancers: A Systematic Review

**DOI:** 10.3390/cancers14153585

**Published:** 2022-07-22

**Authors:** Boon Fei Tan, Wei Chang Colin Tan, Fu Qiang Wang, Matt Lechner, Volker Hans Schartinger, Daniel Shao Weng Tan, Kelvin Siu Hoong Loke, Wen Long Nei

**Affiliations:** 1Division of Radiation Oncology, National Cancer Centre, Singapore 169610, Singapore; boonfei.tan@mohh.com.sg (B.F.T.); wang.fu.qiang@singhealth.com.sg (F.Q.W.); 2Yong Loo Lin School of Medicine, National University of Singapore, Singapore 117597, Singapore; colin.tan.wc@mohh.com.sg; 3UCL Cancer Institute and Division of Surgery and Interventional Science, University College London, London WC1E 6DD, UK; m.lechner@ucl.ac.uk; 4Department of ENT, Barts Health NHS Trust, London E1 1FR, UK; 5Department of Otorhinolaryngology, Medical University of Innsbruck, 6020 Innsbruck, Austria; volker.schartinger@i-med.ac.at; 6Division of Medical Oncology, National Cancer Centre, Singapore 169610, Singapore; daniel.tan.s.w@singhealth.com.sg; 7Department of Nuclear Medicine and Molecular Imaging, Division of Radiological Sciences, Singapore General Hospital, Singapore 169608, Singapore; kelvin.loke.s.h@singhealth.com.sg

**Keywords:** adenoid cystic carcinoma, salivary gland cancers, prostate specific membrane antigen, positron emission tomography, theranostics

## Abstract

**Simple Summary:**

Adenoid cystic carcinoma (ACC) and other salivary gland cancers (SGC) are rare conditions with limited treatment options when they recur or spread to other parts of the body. There is increasing interest in the interaction of radioactive labeled proteins 68Gallium- Prostate Specific Membrane Antigen (68Ga-PSMA) with their corresponding receptors on tumor cells (PSMA receptor) which can be detected on scans. This innovation has created diagnostic and therapeutic progress in management of metastatic prostate cancer. These interactions are also found in SGCs though studies are currently limited. Our systematic review aims to collate available published scientific information on this technology to better inform its potential use, pitfalls and its future directions as a diagnostic and therapeutic option in SGCs. We concluded that the 68Ga-PSMA scans can be useful in detecting ACC and SGC not detected on standard radioimaging and that small studies have shown the therapeutic potential of this innovation in advanced or metastatic ACC and SGC.

**Abstract:**

Adenoid cystic carcinoma (ACC) and other salivary gland cancers (SGCs) are rare tumors where application of prostate specific membrane antigen (PSMA) positron emission tomography (PET) and PSMA radioligand therapy have yet to be studied extensively. This review explores the role of PSMA PET imaging and therapy as a theranostic tool for ACC and other SGCs based on current literature. A comprehensive literature search on PubMed and Embase was performed. All relevant studies containing information on PSMA PET imaging in ACC and SGC were included. Ten studies (one prospective, three retrospective, five case reports and one review paper) were included. For ACC, the mean maximum standardized uptake value (SUVmax) for local recurrence and distant metastases ranged from 2.41 to 13.8 and 2.04 to 14.9, respectively. In SGC, the meanSUVmax ranged from 1.2–12.50. Most studies observed PSMA expression positivity on immunohistochemistry (IHC) when there was PSMA PET uptake. PSMA PET was able to detect lesions not detected on standard imaging. Despite the small number of studies and wide intra-patient and inter-tumor variation of PSMA uptake in ACC and SGC, 68Gallium (68Ga)-PSMA PET has promising prospects as a diagnostic and radioligand therapeutic option. Further studies to answer the various theranostics considerations are required to guide its use in the real-world setting.

## 1. Introduction

Salivary gland cancers (SGCs) are rare malignancies with an annual incidence rate of 0.4–2.6 cases per 100,000 [1]. The World Health Organization (WHO) classification of primary SGCs is based almost exclusively on histomorphology and it comprises up to 20 distinct histopathologic entities [1]. The common histologic subtypes include mucoepidermoid carcinoma, adenoid cystic carcinoma (ACC), acinic cell carcinoma, carcinoma ex pleomorphic adenoma, etc. ACC accounts for 10% of all salivary gland neoplasms and only 1% of all malignant head and neck tumors [2,3,4]. Though slow growing with a low incidence of nodal spread, local and distant recurrence of ACC is relatively common after primary resection due to the propensity for perineural invasion, occult extension beyond surgical margins and hematogenous spread at early stages of the carcinoma [5,6,7]. ACCs occur more frequently in the minor salivary glands which are mainly located in the mucosa of the palate, lips, buccal mucosa, tongue and floor of the mouth. ACC can also arise from other sites of the head and neck such as the tongue, paranasal sinuses, palate, nasopharynx, lacrimal glands, external auditory canal and various secretory glands located in other tissues such as the tracheobronchial tree, esophagus, breast, lungs, prostate, uterine cervix and vulva [8,9]. The most common sites of metastases for ACC are in the lungs, bone and liver.

The diagnosis of SGC can be challenging. Common salivary glands such as the parotid and submandibular glands can be adequately visualized using current clinical imaging modalities such as computed tomography (CT), magnetic resonance imaging (MRI) and fluorodeoxyglucose positron emission tomography/computed tomography (FDG PET/CT) but smaller salivary glands can be extremely challenging to detect [10]. There is thus a need for more effective imaging modalities for salivary gland cancers such as functional imaging.

In recent years, there has been increasing interest in the use of radioligand imaging including prostate specific membrane antigen (PSMA) PET particularly in cancer staging and tumor delineation in radiation oncology. The use of PSMA PET in prostate cancer has been extensively studied. Meanwhile, in SGCs, PSMA PET also has potential as a diagnostic tool though currently this is in the experimental phase. PSMA is a transmembrane glycoprotein of the prostate secretory acinar epithelium that is upregulated in prostate cancer [11]. It is only metabolically active in its homodimeric form and contains a large extracellular domain which enables it to be targeted using antibodies or small-molecule antagonists which are transported into the cell. Within the cell, an internalization motif increases the deposition of conjugated radiometals into the cells thus improving imaging and therapeutic efficacy [12,13,14,15,16]. Whilst PSMA has largely been studied in the arena of prostate cancer, increasing studies have found the presence of PSMA expression in other solid tumors or their neovasculature, rendering it a misnomer [17]. It has been detected in normal body tissues such as the kidneys, in common malignancies such as breast, lung, colorectal and bladder cancers as well as in rare malignancies such as salivary gland, gall bladder and pancreatic cancers [17,18,19,20]. PSMA specific ligands that can be labeled with radioisotopes, such as 18-Fluorine (18F) and 68-Gallium (68Ga), have formed the basis of nuclear imaging in prostate cancer and the latter has been found to visualize substantially more tumor lesions than other modalities [21]. ACC and other SGCs have also been shown to have PSMA expression and uptake of 68Ga-PSMA, making 68Ga-PSMA PET a promising diagnostic tool [19,20,22,23].

The progress in studies of PSMA radioligands has led further to the innovation of radioligand therapy which currently sees its role in metastatic prostate cancer with a recent landmark trial (VISION) showing good response with minimal toxicities [24]. Meanwhile, for early or locally advanced disease in ACC, the standard treatment includes surgery and/or radiotherapy with good prognosis but when recurrent or metastatic to cause symptoms or when they rapidly progress, though they are generally slow growing tumors, treatment strategies are limited by resectability or radiation dose constraints. Treatment options are often limited to chemotherapy and clinical trials with novel agents such as tyrosine kinase inhibitors [25]. Even so, the outcomes are modest at best with median overall survival with best supportive care of 5 months [26,27]. Drawing on the experience of 177Lutetium (177Lu)–PSMA-617 in the treatment of metastatic castration resistant prostate cancer (mCRPC), this adds treatment possibilities for ACC and other SGCs. Using β-emitting radioligand 177Lu attached to the PSMA specific ligand (Figure 1), 177Lu-PSMA-617 radioligand treatment has been used to treat mCRPCs that have failed conventional treatment options with relatively good outcomes and low treatment toxicity [28,29,30,31]. The VISION trial is one of the most recent significant phase III trials which looked at the outcomes of progressive PSMA-positive mCRPC that received 177Lu-PSMA-617 in addition to best supportive/best standard of care versus best supportive/best standard of care alone. It showed significantly good outcomes with 177Lu-PSMA-617 treatment [24,32]. This reflects an emerging field in oncology called theranostics whereby the same molecular ligand is used for both diagnostic and therapeutic purposes and holds potential in the treatment of recurrent or metastatic ACC and other SGCs.

This systematic review aims to explore the possible role of PSMA PET imaging as a diagnostic tool for ACC and other SGCs based on current literature. This will shed light on the potential therapeutic use in this group of conditions.

## 2. Materials and Methods

### 2.1. Search Strategy

The study was performed according to the Preferred Reporting Items for Systematic Reviews and Meta-Analyses (PRISMA) guidelines [33]. Two authors (B.F. Tan and W.C.C. Tan) independently and systematically searched PubMed and Embase for all relevant studies published from inception to 31 December 2021. Our search strategy is based on a combination of the following terms: (a) “PSMA” AND (b) “salivary gland cancers” AND/OR (c) “adenoid cystic carcinoma” and all the relevant variants of the respective terms. Studies were included if they contained information on PSMA PET imaging in ACC and SGC. Only articles in the English language were included. This included original research articles, conference abstracts and editorials. Articles were excluded if they had an irrelevant topic, wrong tumor type, wrong type of imaging and wrong analysis. B.F. Tan and W.C.C. Tan independently reviewed the titles and abstracts of the retrieved articles followed by the full-text version of the relevant articles to determine the eligibility. The reference lists of identified studies were reviewed for relevant articles not picked up from the search strategy. Incongruities in selection were resolved by consensus after comprehensive discussion.

### 2.2. Data Extraction and Analysis

Two reviewers further evaluated the search results independently and derived the data independently using a standardized collection form. Data extracted include the details of the study (author names, type of study, year of study, country of origin), patient (number, age, gender, Eastern Cooperative Oncology Group (ECOG) status), tumor (tumor type, TNM staging, sites of metastases, treatment) and PSMA uptake (PSMA peptide used, sites of PSMA uptake, maximum standard uptake value (SUVmax) in the primary and secondary salivary gland malignancies, PSMA receptor expression on immunohistochemistry (IHC)).

The data were analyzed and the overall weighted mean SUVmax and the weighted mean SUVmax for each group of involved subsites was calculated. The data to calculate this were only derived from four papers as only these papers provided robust reporting on the SUVmax [19,23,34,35]. Information on methods of defining abnormal PSMA PET uptake and whether PSMA PET led to a change in clinical management was also gathered.

#### 2.2.1. Overall Mean SUVmax

From the paper by van Boxtel et al., for each of the 24 patients (15 with ACC, 9 with salivary duct carcinoma (SDC)), the authors reported the mean SUVmax for each involved subsite [23]. The number of lesions varied from subsite to subsite and patient to patient. Hence, to achieve a more uniform way of measurement, the overall weighted mean SUVmax was derived for each patient (see Appendix A).

In the paper written by Klein Nulent et al. (2017), for each of the nine patients, they reported the SUVmax for local recurrence and for each distant metastasis [19]. They, however, did not state the number of lesions for each subsite. Hence, it was not possible to derive the weighted mean SUVmax. Using the information from this paper, the mean SUVmax per patient (see Appendix A) was derived. In the remaining papers, the number of metastatic lesions per site was not clearly stated and, so, there was no need to calculate the weighted mean SUVmax here (see Appendix A). The final overall mean SUVmax was calculated by adding up the weighted mean SUVmax for the paper by van Boxtel et al. with the mean SUVmax for the other three papers and finding the overall mean SUVmax for the 29 patients with ACC (see Appendix A). The same was repeated for obtaining the overall mean SUVmax for the 12 patients with other SGCs (see Appendix A).

#### 2.2.2. Mean SUVmax per Subsite

To determine the mean SUVmax for metastatic lesions, the mean SUVmax values for each subsite were obtained and the weighted mean SUVmax calculated (see Appendix A).

### 2.3. Quality Assessment

The included studies were independently assessed for risk of bias using the Quality Assessment of Diagnostic Accuracy Studies (QUADAS)-2 tool [36]. The tool consists of four key domains to assess for risk of bias and applicability concerns, and these domains are patient selection, index test, reference standard and “flow and timing”, which refers to the flow of patients through the study and timing of the index test(s) and reference standard. The domains are reviewed and determined to be at low, high or unclear risk of bias. If a study is judged as “low” on all domains relating to bias or applicability, then it is appropriate to have an overall judgment of “low risk of bias” or “low concern regarding applicability” for that study. If a study is judged “high” or “unclear” on one or more domains, then it may be judged “at risk of bias” or as having “concerns regarding applicability”.

## 3. Results

### 3.1. Study Selection

Figure 2 depicts the PRISMA flow chart of the literature search and article selection. Here, 2117 studies were identified through database searches, and 563 duplicates were removed, leaving 1554 studies. After assessment of title and abstract, 1513 articles were excluded for irrelevant topic, wrong tumor type, wrong type of imaging and wrong analysis. The full text and references of the remaining 41 articles were analyzed for eligibility, where 10 articles were included in the review and no additional study was found within the references.

### 3.2. Study Characteristics

The articles identified include one prospective study, three retrospective studies, five case reports and one review paper. Out of the four prospective and retrospective studies, three of them were published studies while the other one was a conference abstract. Most studies looked at ACC whilst two papers included other SGCs. A summary of the findings from the review can be found in Table 1.

#### 3.2.1. Setting of PSMA PET Use

Only two studies, both case reports, investigated PSMA PET in a primary setting which had advanced loco-regional involvement [35,41]. For both these papers, the ACC was already histologically confirmed and the PSMA PET scan was performed after standard staging imaging. ACC was reported as an incidental finding in one case report [37]. The PSMA PET was performed on suspicion of prostate cancer in view of a raised prostate specific antigen (PSA) but the biopsy of the avid bone metastasis confirmed metastatic ACC. All other ACC/SGCs reported were neither suspicious findings nor secondary findings. In the other studies, patients were referred for PSMA PET imaging after they had failed conventional treatment for their recurrent/metastatic ACC and consequently the authors only reported the SUV readings for the recurrent/metastatic lesions. In these reports, there were no mentions of whether these patients had prior PSMA PET for their primary disease.

#### 3.2.2. Defining Abnormal Uptake on PSMA PET

Only three of ten papers elaborated on the definition of abnormal uptake on the PSMA PET [19,23,34]. The authors measured the SUVmax and SUVmean in various organs/structures including the lacrimal gland, major salivary glands, liver and mediastinal blood pool in order to assess the normal biodistribution. Tumor/organ ratios were calculated. A tumor/liver ratio of >1 was regarded as relevant PSMA ligand uptake. In another case report by König et al. [35], the patient’s primary ACC in the maxillary sinus was biopsy-confirmed and the extent of PSMA PET uptake corresponded to the extent of disease seen on the staging MRI scan. The remaining included studies looked at locally recurrent and metastatic SGCs that were not within salivary glands. In the article by de Keizer et al. [37], biopsy of these metastatic avid regions confirmed malignant lesions. Similarly, in the study by van Boxtel et al. [23], some of these avid lesions were known histologically confirmed metastatic ACCs. In the case reports by Lutje et al., Has Simsek et al. and Dhiantravan et al., a corresponding MRI or CT scan was carried out which showed the extent of the metastatic lesions [20,39,41]. Furthermore, Klein Nulent et al. and Dhiantravan et al. performed serial PSMA PET scans which showed changes in SUVmax post treatment in the areas of uptake, indicating that they are unlikely to be physiologically normal tissue [34,41]. All included studies except that by Kiess et al. published the relevant PSMA PET images [38].

#### 3.2.3. Change in Management following PSMA PET

In eight papers, there were reported changes in clinical management for the patients following the findings on PSMA PET. In three of them, the PSMA PET picked up new metastatic lesions not seen on standard imaging and this led to further palliative treatment [19,20,23]. In two of the papers, the PSMA PET showed more extensive primary loco-regional disease involvement than seen on MRI scan, resulting in a change in radiotherapy target volume delineation over an area that would otherwise not be considered in the treatment plans [35,41]. Additionally, for three papers, the patients included were referred for PSMA PET to investigate if radioligand therapy would be appropriate [34,39,40]. For these patients that showed uptake on the PET, they went on to receive palliative radioligand therapy.

### 3.3. Mean SUVmax for ACC

There is a wide range of SUVmax seen for primary loco-regional, local recurrences and in the different metastatic sites. Four papers provided detailed information regarding the SUVmax corresponding to the different sites of disease [19,23,34,35]. From these four papers, the overall mean weighted SUVmax, weighted mean SUVmax for local recurrence and weighted mean SUVmax for metastases for adenoid cystic carcinoma were 7.21 (min–max range 2.04–23.35), 6.33 (min–max range 2.41–13.8) and 6.82 (min–max range 2.04–14.9), respectively (Table 2) (Appendix A). Figure 3 and Appendix A show the overall weighted mean SUVmax per metastatic subsite for ACC. The highest mean SUVmax was 23.35 in the maxillary sinus and the lowest was 6.93 in the neck metastases.

### 3.4. Mean SUVmax for Other SGCs

Meanwhile, the overall mean weighted SUVmax, weighted mean SUVmax for local recurrence and weighted mean SUVmax for metastases for other SGCs were 5.99 (min–max range 1.2–12.50), 10.57 (min–max range 4.0–16.8) and 6.06 (min–max range 1.43–14.27), respectively (Appendix A). Appendix A shows the overall weighted mean SUVmax per metastatic subsite for other SGCs. The highest mean SUVmax was 9.19 in the bone metastases whilst the lowest was 2.37 in subcutaneous metastases.

### 3.5. Quality Assessment

Results of the quality evaluation of the studies can be seen in Table 3.

### 3.6. Ongoing Clinical Trials

We also summarized the ongoing clinical trials related to PSMA theranostics in SGC as of 31 December 2021 to demonstrate the current progress in this field. A group of researchers in the Netherlands is looking to investigate if androgen deprivation therapy can increase the uptake of 68Ga-PSMA in patients with salivary duct cancers (SDCs), as has previously been demonstrated in prostate cancer (NCT04214353). The same group also aims to evaluate the safety and efficacy of 177Lu-PSMA radioligand therapy in patients with recurrent or metastatic ACC and SDC with PSMA ligand uptake (NCT04291300). Lastly, we noted an ongoing trial based in China which aims to evaluate 68Ga-PSMA-617 uptake in local recurrent or metastatic ACC in comparison with 18F-FDG uptake in the same patients and assess the feasibility of 177Lu-EB-PSMA-617 treatment in patients with advanced ACC (NCT04801264). All three trials are actively recruiting, and a summary can be found in Table 4.

## 4. Discussion

This is the first systematic review looking at the use of PSMA PET imaging and therapy in ACC and other SGCs. In this review, we see that PSMA PET has a promising role in diagnostic imaging and radioligand therapy for recurrent and metastatic SGC. However, the overall small number of studies comprising mainly case reports and retrospective studies may limit its interpretation and use in real-world settings currently, highlighting the unmet need for more comprehensive prospective studies.

### 4.1. PSMA PET as a Diagnostic Tool for SGC

This review shows that studies mainly described the use of PSMA PET in the recurrent/metastatic setting with 93% and 40% of patients having PSMA ligand uptake in recurrent/metastatic ACC and SDC, respectively [23]. Only two case reports described PSMA PET in the primary setting where the disease was advanced with loco-regional involvement [35,41].

Whilst the utilization of PSMA PET may not be recommended as standard imaging in view of accessibility, cost and the need for further confirmatory studies, as seen in this review, it can be considered in certain circumstances. PSMA PET can be used when there is discrepancy between standard imaging and clinical features. In one of the reports, PSMA PET was able to detect symptomatic bone metastasis not seen on CT scan. The patient went on to receive radioligand therapy with resulting symptomatic relief [39]. In three other studies, PSMA PET picked up new metastatic lesions not seen on standard imaging and this led to further palliative treatment [19,20,23]. Another diagnostic utility is when indeterminate findings on conventional imaging need to be better characterized for better tumor delineation for radiotherapy planning or if it would lead to a change in treatment regime. For example, two papers reported that the PSMA PET showed more extensive primary loco-regional disease involvement than seen on MRI scan, resulting in a change in radiotherapy target volume delineation over an area that would otherwise not be considered in the treatment plans [35,41]. PSMA PET imaging may also be preferred over the more frequently used FDG PET-CT as a diagnostic tool in ACC as it commonly invades the perineural structures which normally shows physiological uptake on the FDG PET-CT, making it difficult to interpret and may be undiagnosed. Hence, in patients with suspected perineural invasion or where the disease requires better delineation such as in the head and neck region, PSMA PET may be a preferred tool over FDG PET. Thirdly, patients can be considered for PSMA PET when they have failed conventional treatment for recurrent/metastatic disease and are considered for radioligand therapy. In the three papers that report this, patients that showed uptake on the PSMA PET went on to receive palliative radioligand therapy [34,39,40]. Other potential uses that need further validation include the monitoring of disease response with serial PSMA PET and the use of PSMA PET in the primary setting. The latter may be challenging given the high physiological uptake of salivary glands [35].

Whilst it appears that PSMA PET is a promising diagnostic tool for SGC, this review identified several pitfalls and aspects that need to be ironed out before it can be uniformly and reliably utilized. Firstly, normal salivary glands are known to have physiological uptake of PSMA radioligand. There has yet to be a study to compare the biometric measurement of PSMA uptake between SGC and normal salivary gland tissues to better guide the diagnostic interpretation [42]. Secondly, the interpretation of positive PSMA uptake is poorly reported. Only three papers elaborated on the method of defining abnormal PSMA uptake and less than half of the studies reported the SUV values of PSMA uptake. In one of the more robust studies [23], positive PSMA uptake was defined as an SUV tumor to liver ratio of more than one, rather than using the SUV value alone. Even when the SUV values were adequately reported, there appears to be a wide intra-patient and inter-tumor variation in PSMA uptake. In some, the reported tumor SUVmax values were within the ranges reported for normal salivary glands [42]. For ACC, the SUVmax for local recurrence ranged widely from 2.41–13.8 and that of distant metastasis was 2.04–14.9 In other SGCs, the SUVmax ranged from 4.0–16.8 for local recurrence and 1.43–14.27 for distant metastasis. Furthermore, the response criteria when using PSMA PET as a diagnostic tool in SGC need to be defined. In prostate cancer, there are additional biomarkers such as PSMA expression and serum PSA levels to correlate PSMA PET response findings. However, in SGC, there is currently no correlating biomarker to validate response on the PSMA PET. In a study by Klein Nulent et al., they defined progression of disease as an increase of ≥20% in SUVmax or tumor volume or when new lesion(s) were discovered, complete response as when all tumor localizations disappeared, partial response as a ≥30% decrease in SUVmax and stable disease as when there was neither partial nor progressive disease [34]. Hence, clinicians need to be aware of these when interpreting PSMA images and a more standardized method of reporting positive PSMA uptake needs to be implemented. Another consideration is the use of PSMA expression on immunohistochemistry (IHC) as a tool for patient selection for PSMA radioimaging. Most studies were consistent with PSMA expression positivity on IHC when there was PSMA PET/CT uptake [19,23,38]. On the other hand, van Boxtel et al. also showed that IHC PSMA expression did not reliably predict PSMA uptake as even those with negative expression had positive PSMA uptake [23]. This is in contrast to data in prostate cancer, where a strong correlation was found between IHC PSMA expression and PSMA ligand uptake, SUV values and disease aggressiveness [43,44]. With only five of 10 studies reporting on PSMA IHC expression, one has to interpret this information with caution. We may conclude that PSMA expression should not be a determining factor in patient selection for PSMA PET imaging for SGC though further prospective studies should be performed to confirm this.

### 4.2. PSMA Radioligand Therapy for SGC

With about 93% PSMA uptake in ACC and 40% PSMA uptake in other SGCs, there are promising prospects for the use of radioligand therapy in SGC and this can be in spite of negative PSMA expression [23]. This combined use of a radiopharmaceutical to both diagnose and deliver therapy to treat a tumor is termed theranostics. This novel option is particularly crucial as the treatment options for SGC, especially when recurrent or metastatic, are limited by lack of effective treatment options.

The experience of radioligand therapy in mCRPC can potentially be translated to ACC and other SGCs. 177Lu-PSMA selectively binds to PSMA and delivers short range beta radiation. In the recent VISION trial comparing 177Lu-PSMA-617 and standard care versus standard care alone, there was a good response rate with significantly improved progression free survival (8.7 months vs. 3.4 months, HR 0.40) and overall survival (15.4 months vs. 11.3 months, HR 0.62) with 15% more adverse events [24]. The common adverse events with 177Lu-PSMA-617 therapy were fatigue (43.1%), xerostomia (38.8%) and nausea (35.3%). The authors identified five adverse events that lead to 177Lu-PSMA-617 related death: pancytopenia, bone marrow failure, subdural hematoma and intracranial hemorrhage. In this trial, participants received intravenous (IV) infusions of 177Lu-PSMA-617 at a dose of 7.4GBq (200mCi) once every 6 weeks for four cycles. In the TheraP trial comparing 177Lu-PSMA-617 to cabazitaxel in mCRPC, there were 29% more PSA responders in the 177Lu-PSMA-617 arm with 15% more progression free survival at 12 months with fewer grade 3 or 4 adverse events, rendering it a new effective class of therapy and potential alternative to chemotherapy [31,45].

In SGC, evidence on 177Lu-PSMA therapy is based mainly on the experience of the case report by Has Simsek et al. and case series by Klein Nulent et al. [34,39]. They reported that 177Lu-PSMA radioligand treatment shows potential as a palliative treatment in both locally recurrent and metastatic ACC whereby participants showed radiological and clinical response. Has Simsek et al. demonstrated in their case report that 177Lu-PSMA therapy was a safe therapeutic option for pain relief in their patient who had metastatic ACC of the parotids and had failed six cycles of chemotherapy [39]. A more recent case series by Klein Nulent et al. further demonstrated feasibility of 177Lu-PSMA-617 therapy for recurrent or metastatic SGC [34]. The treatment, similar to the VISION trial, aimed at four cycles of IV administration of 6–7.4GBq 177Lu-PSMA-617 with an interval of 6–8 weeks. Interestingly, one of six patients had stable disease post four cycles of 177Lu-PSMA-617 therapy whilst another patient saw response both radiologically (SUVmax decrease of 30%) and clinically for up to 10 months after the start of treatment. In three other patients, there was radiological disease progression, though, in two of them, clinical symptoms improved. These three patients did not receive the full four cycles due to disease progression. In the two patients with clinical improvement despite radiological progression, it is also not clear whether clinical improvement is purely due to 177Lu-PSMA-617 therapy or other pain relief treatments. One patient received only one cycle due to demotivation from side effects and one patient was found to have grade 3 thrombocytopenia (though the authors commented that this was likely from bone marrow related disease progression rather than treatment toxicity). Nevertheless, the overall toxicities appear tolerable with grade 1-2 fatigue and xerostomia being more common. There was no correlation between pre-treatment SUVmax and tumor tissue PSMA expression and the percentage of PSMA expression on IHC did not appear to influence the response. The patients who had radiological response had SUVmax 3.5 to 6.5 whilst those without response had SUVmax 7–12.5. Currently, the use of radioligand therapy has not been explored in the primary local setting. Nevertheless, radioligand therapy in locally recurrent/metastatic ACC is still in its exploratory phase with weak evidence in the form of a case report and retrospective case series. More detailed and prospective studies are required to improve on the evidence on its use. However, given the rarity of this condition, it may be difficult to perform large scale studies.

There are several considerations yet to be explored regarding radioligand therapy in ACC. Unlike mCRPC, the wide intra-patient and inter-metastatic variation in PSMA uptake in SGC makes patient selection for the radioligand treatment challenging. Currently, there are no data available to show if radioligand therapy is effective even when there is no PSMA uptake. From the perspective of IHC PSMA expression, its expression in primary tumor could not predict expression levels in disease recurrence or metastasis although a tendency to increase and decrease was observed, respectively [18]. This is in contrast with mCRPC where high IHC PSMA expression is correlated with disease recurrence and other negative prognostic factors [46]. This may be an implication when considering these patients for radioligand treatment. On the other hand, van Boxtel et al. showed that even those with negative IHC PSMA expression had PSMA uptake [23]. Further studies on the effect of radioligand therapy on SGCs that do not express IHC PSMA or PSMA uptake are needed to address these issues. A second consideration is the appropriate timing of radioligand treatment. In prostate cancer, there is a correlation of increased PSMA expression with increased stage, grade and PSA, indicating that radioligand therapy may be more beneficial later in the disease [47]. Even so, the TheraP trial showed better outcomes with radioligand therapy compared to cabazitaxel in mCRPC, highlighting the potential of introducing radioligand therapy earlier in the treatment algorithm [31]. In the SGC experience, the only available study of radioligand therapy in this group of patients saw the treatment introduced as a last resort [34]. In a study by van Boxtel et al., a negative trend in SGC PSMA uptake between the time of diagnosis to the time of the PET CT scan was observed [23]. This observation highlights that the timing of radioligand therapy can be explored in the future to help determine the possibility of introducing radioligand therapy as an option earlier in the treatment algorithm, such as alongside novel systemic therapies, rather than as a last resort. Thirdly, unlike SGC, metastasis to the brain in prostate cancer is uncommon. As such, we do not know the effectiveness of radioligand therapy for cerebral metastasis and the associated toxicities when utilized in SGC.

The choice of radioligand should also be paid attention to. Alpha emitting radioligands have a soft tissue range of a few micrometers with very high linear energy transfer and, thus, highly targeted toxicity effect. Beta emitting radioligands have a soft tissue range of a few millimeters but a lower energy transfer. In the phase 3 ALSYMPCA trial, alpha emitting radioligand Radium-223 was used as targeted radioligand therapy whilst the VISION trial investigated the use of beta emitting 177Lu-PSMA-617 in mCRPC, both of which showed improved overall survival and tolerability [24,47]. One of the major dose-limiting toxicities is xerostomia (Grade 1 in 87%) [45]. Unlike other tumor sites, an important consideration when used to treat SGC is the increased side effect of xerostomia which is expected to be worst in these patients as most would have undergone surgical excision and/or radiotherapy and be debilitated from the post-treatment effects. In this situation, alpha emitting radioligand may be less preferred as it may induce xerostomia more often compared to beta emitting radioligand. All these unanswered aspects specific to radioligand therapy in SGC warrant a phase II study.

Along with the aforementioned gaps and limitations, a significant limitation in our review is the lack of robust studies and uniform information presentation, hindering the conclusions that can be drawn. About 50% of the studies were case reports or conference abstracts which generally have a paucity of information when it comes to patient details such as ECOG status and TNM staging of the tumor. Most of the studies were conducted in the European countries of the Netherlands and Germany. The SGC subtypes reported in this review were predominantly adenoid cystic carcinoma with some cases of salivary duct carcinoma. Lack of information about the studies also leads to great difficulty in assessing for risk of bias. Future studies should include more comprehensive demographic, disease, diagnostic and treatment information such as symptoms, disease staging, patient’s performance status, previous treatment and radiopeptide used and SUV measurements.

## 5. Conclusions

In conclusion, whilst the SUV measurements of PSMA PET in SGC are highly variable, there is a potential role of PSMA PET as a diagnostic tool, especially when conventional imaging is inconclusive or does not correlate with clinical findings. Whilst limited information is available, generally there is PSMA expression on IHC in SGC, though more studies are required to identify the correlation with PSMA uptake on PET. The use of PSMA ligand has the potential to be a form of theranostics whereby it can be used as a diagnostic and therapeutic tool in the management of SGC, opening up new treatment options for this disease entity which has been challenging to treat when recurrent or metastatic and where limited effective systemic therapy exists.

## Figures and Tables

**Figure 1 cancers-14-03585-f001:**
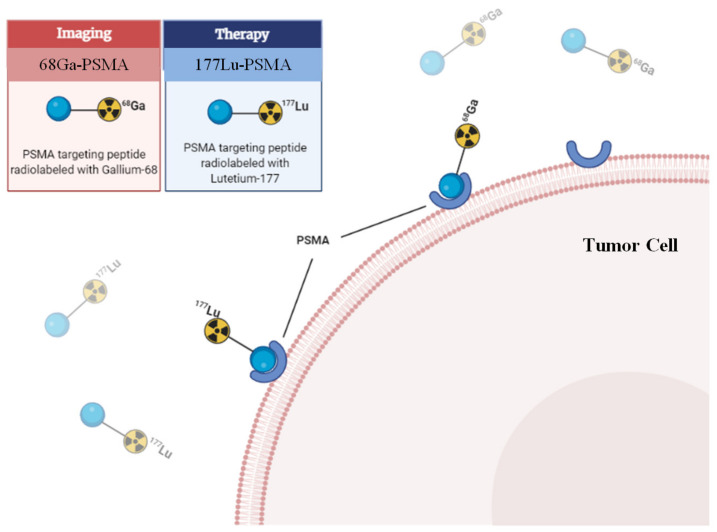
PSMA radioligand and receptor interaction. 68Ga: 68Gallium; 177Lu: 177Lutetium.

**Figure 2 cancers-14-03585-f002:**
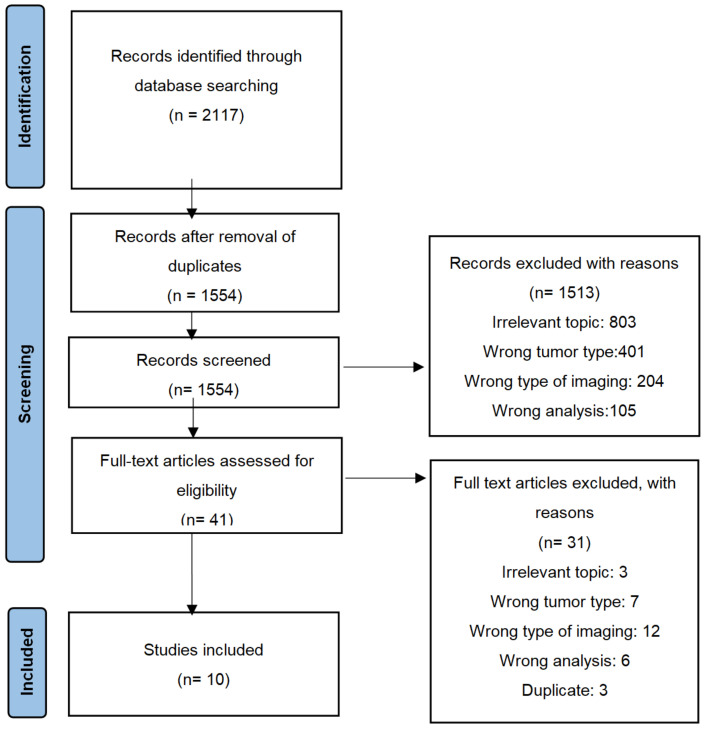
PRISMA flow chart.

**Figure 3 cancers-14-03585-f003:**
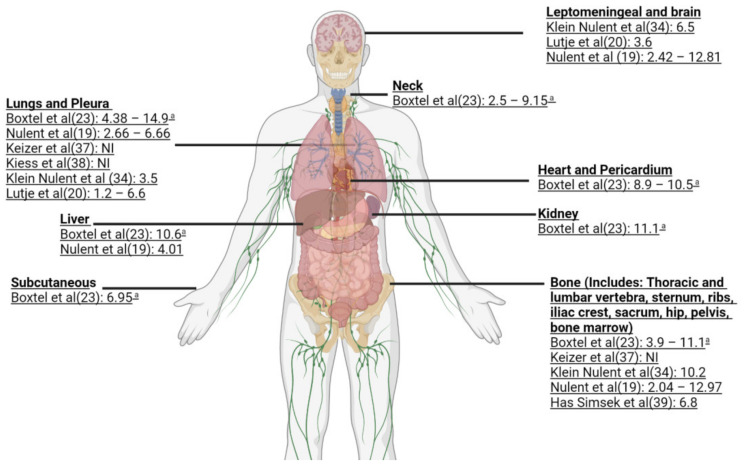
Sites of metastases of ACC with reported mean SUVmax. The image contains reference citations indicated within parentheses [19,20,23,34,37,38,39]. ^a^-Range of mean SUVmax; NI-No information.

**Table 1 cancers-14-03585-t001:** Characteristics of studies.

Study Author (Year of Publication)	Location	Study Type	No. of Patients	Median Age/Range	Tumor Type (Setting)	Isotope	PSMA Peptide	Sites of Metastases with PSMA Uptake	Mean SUVmax	Range of SUVmax	Tumor/Liver Ratio (SD, Range)	PSMA Expression Findings
**de Keizer et al. (2017)** [37]	Netherlands	Case report	1	NI	ACC (metastatic)	68Ga	NI	Lung, bone	NI	NI	NI	High
**Lutje et al. (2016)** [20]	Germany	Case report	1	58	ACC (metastatic)	68Ga	HBED-CC	Lung, brain	NI	Distant metastases: 1.2–6.6	NI	NI
**Kiess et al. (2018)** [38]	USA	Abstract	3	NI	ACC (metastatic)	18F	DCPyL	Cervical lymph nodes, lung	NI	Metastatic: 1.0–6.3	NI	NI
**König et al. (2017)** [35]	Germany	Case report	1	58	ACC (primary, loco-regional)	68Ga	NI	Right maxillary sinus	23.25	NI	NI	IHC staining for PSMA showed low cytoplasmatic positivity in approximately 5% of ACC cells
**Klein Nulent et al. (2017)** [19]	Netherlands	Retrospective	9	66/(31–76)	ACC (locally recurrent and metastatic)	68Ga	HBED-CC	Leptomeningeal, lungs, iliac crest, intracranial, vertebra	Local recurrence: 3.63 Distant metastases: 5.35	Local recurrence: 2.41–7.06 Distant metastases: 2.04–12.97	5 of 9 patients (55.6%) had tumor/liver ratio > 1	All tumors including primary, local recurrent and distant metastases showed PSMA expression on IHC (5–90% expression), IHC PSMA expression has no association with PSMA uptake on PET/CT
**Has Simsek et al. (2019)** [39]	Turkey	Case report	1	48	ACC (metastatic)	68Ga	PSMA-617	Sternum, ribs, vertebra, iliac bone, sacrum, bone marrow	6.8	2.9–14.8	1.71 (0.61, 0.43–2.2)	NI
**De Galiza Barbosa et al. (2020)** [40]	Brazil	Review	1	NI	ACC (metastatic)	68Ga	11-HBED	NI	NI	NI	NI	NI
**Van Boxtel et al. (2020)** [23]	Netherlands	Prospective	25	ACC: 58 (44–76) SDC: 69.5 (55–79)	ACC/SDC (locally recurrent and metastatic)	68Ga	HBED-CC	Heart, lung, brain, kidney, subcutaneous neck, bone, pleura, distant lymph node, pericardium	NI	ACC: 1.1–30.2 SDC: 0.3–25.9	ACC: tumor/liver-ratio >1 in 13 out of 14 evaluable patients (93%). SDC: tumor/liver-ratio >1 in 4 out of 10 patients (40%)	In the ACC patients, the median percentage of PSMA-positive tumor cells was 7.5% (range 0–90%) in the resected primary tumors and 5% (range 0–80%) in 11 biopsies from metastases. IHC PSMA expression in the tumor-associated neovasculature was negative in all ACC patients. In SDC patients, median percentage of PSMA-positive tumor cells was 0% (range 0–50%) with 5 out 9 patients showing no PSMA expression in the resected primary tumors, and 0% in one biopsied metastatic lesion. IHC PSMA expression in the tumor-associated neovasculature was positive in 8 of 9 evaluable SDC patients
**Dhiantravan et al. (2021)** [41]	Australia	Case report	1	56	ACC (primary, loco-regional)	NI	NI	Intracranial	NI	NI	NI	NI
**Klein Nulent et al. (2021)** [34]	Netherlands	Retrospective study	6	40	ACC/SGC^a^ (locally recurrent and metastatic)	68Ga	NI	Lung, intracranial, pelvis, parapharyngeal	8.23	3.5–12.5	NI	PSMA expression on IHC showed similar expression patterns within the neoplastic cells ranging from 5–95% (5–30% for ACC); no correlation between pre-treatment SUVmax and tumor tissue PSMA expression; 2 patients with the highest PSMA expression on IHC showed good initial clinical response

ACC: adenoid cystic carcinoma; DCPyL: 2-(3-{1-carboxy-5-[(6-18F-fluoro-pyridine-3-carbonyl)-amino]-pentyl}-ureido)-pentanedioic acid; IHC: immunohistochemistry; HBED-CC: N,N′-bis-[2-hydroxy-5-(carboxyethyl)benzyl]ethylenediamine-N,N′-diacetic acid; 18F: 18-Fluorine; NI: no information; PET/CT: positron emission tomography/computed tomography; PSMA: Prostate Specific Membrane Antigen; SDC: salivary duct carcinoma; SGC: salivary gland cancer; SUVmax: maximum standardised uptake value; 68Ga: 68-Gallium. ^a^ SGC in this study included adenocarcinoma not otherwise specified and acinic cell carcinoma.

**Table 2 cancers-14-03585-t002:** Mean values of PSMA PET SUVmax and range of SUVmax for locally recurrent and metastatic ACC and other SGCs.

	ACC		Other SGCs	
Author	Local Recurrence Mean SUVmax (Range)	Metastases Mean SUVmax (Range)	Local Recurrence Mean SUVmax (Range)	Metastases Mean SUVmax (Range)
Lutje et al. (2016) [20]	-	Lung: no mean (1.2–6.6)	-	-
Kiess et al. (2017) [38]	Not reported	Not reported	-	-
König et al. (2017) [35]	-	23.35 (no range)	-	-
Klein Nulent et al. (2017) [19]	3.63 (2.41–7.06)	5.35 (2.04–12.97)	-	-
Van Boxtel et al. (2020) [23]	9.73 (4.8–13.8) ^a^	7.62 (2.5–14.9) ^a^	10.57 (4.0–16.8) ^a^	4.94 (1.43–14.27) ^a^
Klein Nulent et al. (2021) [34]	7.0 (no range)	6.73 (3.5–10.2)	-	11.1 (9.7–12.5)
Overall	6.33 (2.41–13.8) ^a^	6.82 (2.04–14.9) ^a^	10.57 (4.0–16.8) ^a^	6.06 (1.43–14.27) ^a^

^a^, Calculated weighted mean (range of mean SUVmax). The calculations were derived from data published by the relevant study. “-“—parameter not investigated. ACC: adenoid cystic carcinoma; SGC: salivary gland cancer; SUVmax: maximum standardised uptake value.

**Table 3 cancers-14-03585-t003:** Quality assessment of studies using the QUADAS-2 tool.

Study Author (Year of Publication)	Risk of Bias				Applicability Concerns		
	Patient Selection	Index Test	Reference Standard	Flow and Timing	Patient Selection	Index Test	Reference Standard
**De Keizer et al. (2016)** [37]	Low	Low	Unclear	Unclear	Low	Low	Unclear
**Lutje et al. (2016)** [20]	Low	Low	Unclear	Unclear	Low	Low	Unclear
**Kiess et al. (2017)** [38]	Unclear	Unclear	Unclear	Unclear	Low	Unclear	Unclear
**König et al. (2017)** [35]	Low	Low	Low	Low	Low	Low	Low
**Klein Nulent et al. (2017)** [19]	Low	Low	Low	Unclear	Low	Low	Low
**Has Simsek et al. (2019)** [39]	Low	Low	Unclear	Unclear	Low	Low	Unclear
**De Galiza Barbosa et al. (2020)** [40]	Low	Unclear	Unclear	Unclear	Low	Unclear	Unclear
**Van Boxtel et al. (2020)** [23]	Low	Low	Low	Low	Low	Low	Low
**Dhiantravan et al. (2020)** [41]	Low	Low	Low	Low	Low	Low	Low
**Klein Nulent et al. (2021)** [34]	Low	Low	Unclear	Unclear	Low	Low	Unclear

**Table 4 cancers-14-03585-t004:** Summary of ongoing clinical trials on PSMA PET theranostics in salivary gland cancers.

Country	First Posted	Trial Number	Trial Phase	Estimated No. of Patients	Intervention	Radioligands Used	Objective
**The Netherlands**	Jan 2020	NCT04214353	Not applicable ^a^	14	All participants in the study will be injected with 2.0 MBq/kg 68Ga-PSMA for PET/CT imaging, both pre- and post ADT.	68-Ga PSMA	To investigate if androgen deprivation therapy can increase the uptake of 68Ga-PSMA in patients with recurrent or metastatic SDC.
**The Netherlands**	Mar 2020	NCT04291300	Phase II pilot study, single center, two cohorts	10	4 cycles of 7.4 GBq 177Lu-PSMA every 6 weeks.	177-Lu-PSMA-I&T	To evaluate the safety and efficacy of 177Lu-PSMA radioligand therapy in patients with R/M ACC and SDC with PSMA ligand uptake.
**China**	Mar 2021	NCT04801264	Early phase 1	40	All patients diagnosed with ACC underwent 68Ga-PSMA PET/CT scan. If the PET/CT showed high PSMA expression in tumor lesions of some patients, they were intravenously injected with a dose of about 1.85GBq (50 mCi) of 177Lu-EB-PSMA-617 every 8 weeks (±1 week) for a maximum of 3 cycles.	68-Ga PSMA-617 177-Lu-EB-PSMA-617	To evaluate 68Ga-PSMA-617 uptake in local recurrent or metastatic ACC in comparison with 18F-FDG uptake in the same patients, and assess the feasibility of 177Lu-EB-PSMA-617 treatment in patients with advanced ACC.

^a^: Interventional clinical trial, an explorative study. ACC: adenoid cystic carcinoma; ADT: androgren deprivation therapy; PET/CT: positron emission tomography/computed tomography; PSMA: Prostate Specific Membrane Antigen; R/M: recurrent/metastatic; SDC: salivary duct carcinoma; 68Ga: 68-Gallium; 177Lu: 177-Lutetium, 18F-FDG: 18-fluorine-fluorodeoxyglucose

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
