# Peer review of "PSMA PET Imaging and Therapy in Adenoid Cystic Carcinoma and Other Salivary Gland Cancers: A Systematic Review"

_cancers, 2022, doi:10.3390/cancers14153585_

Round 1

Reviewer 1 Report

The idea of this review is scientifically interesting. However, as physiologial uptake of PSMA in the salivary glands is high, the review must focus much clearer on how the cited studies did differentiate malignant disease from physiological uptake. I would appreciate a clear statement in the introduction, that currently PSMA imaging is being evaluated for prostate cancer (as mentioned in the last paragraph) but however imaging of ACC is experimental. Additionally, it remains unclear to me, if the cited studies investigated PSMA in primary or local recurrent tumors. Please clarify.

Additionally, the following points may need to be adressed:

1)    SGC: it appears, that not enough studies with appropriate data are available?

What is included in SGC. Please clarify.

2)    Table 2: please include the used tracers to be able to compare SUVs

3)    Only 5-6 studies report data (SUV) on ACC. Was there a change of clinical managment in these patients?

4)    Table 3 confusing, almost all studies unclear?

5)    Figure 2 interesting, well summarized

6)    Table 2: difference between n/a and not reported?

7)    Please include a paragraph with detailed information about how the authors in the cited papers differentiated between physiologic (high) uptake in the salivary glands and tumors uptake, was data of CT/MRI included? Was ACC suspected in primary place or „just“ a secondary finding?

8) Introduction: lines 65-66, (PSMA) PET imaging has technical/resolution limitations with respect to in plane resolution, I find this statement misleading.

9) in what context was the data on ACC obtained (e.g. did patients have prostate cancer and got PET imaging therefore?). Please comment on that.

10) Discussion: I am missing a paragraph on what we can learn from the 5-6 studies on ACC. Additionally, I would appreciate a more in depth discussion of the 5-6 studies on ACC.

11) It is known from application of 225Ac-PSMA-617 radioligand treatment (and to a minor extend also in 177Lu-PSMA) that xerostomia is a significant side effect. Please at least mention this point in the discussion. What are the expectations of 177Lu-PSMA treatment in ACC and in which setting (local recurrent/metastatic disease)?

12) There is a lot of data in the supplement, is it all necessary? Please revise.

Minor comments: Please use either 177Lu-PSMA insted of 177Lu-PSMA or Lu-177-PSMA instead (also applicable to other tracers/nuclides).

Reviewer 2 Report

The authors conducted a patient-detailed systematic review on PSMA-targeted diagnostics and treatment options in adenoid cystic carcinoma and other salivary gland tumors.

In my opinion, the literature review has been carried out meticulously and the selected papers have been adequately judged against the applicable criteria/risk of bias using QUADAS, despite little details were given by the included case reports.

Figure 3, showing mean SUVmax per metastatic site is well drawn and gives a good summary of the collected data.

However, I have my doubts regarding the inclusion of 'abstracts only', 'posters' and 'conference abstracts'. Especially when they are followed by a published -original research- study. There is a good chance that patients have been counted more than once, or that incomplete data from these -often preliminary reports- have been included. My main request is therefore to consider exclusion of these citations and not to publish the reported data by these abstracts.

Furthermore, it is noted that data is missing/incomplete. The most important data used for the results of this review was extracted from studies by Van Boxtel and Klein Nulent (NL). Have the authors considered contacting these and other cited authors to collect the missing values? This may lead to a better overall comparison, which will certainly benefit the quality of this paper. Because already many data per patient have been provided, I would encourage the authors to rewrite the study into a meta-analysis by adding just the few extra patient-specific details. 

Round 2

Reviewer 2 Report

Major improvements have been made. The systematic review is well readable and gives an excellent overview of the sparse available literature on this topic.